# An Investigation of Spectral Band Selection for Hyperspectral LiDAR Technique

**Hui Shao [1,2]**, **Yuwei Chen [2,3],\***, **Wei Li [3]**, **Changhui Jiang [4]**, **Haohao Wu [3]**, **Jie Chen [1]**, **Banglong Pan [1]** and **Juha Hyyppä [2]**

1   The School of Electronic and Information Engineering, Anhui Jianzhu University, Hefei 230601, China; shaohui@ahjzu.edu.cn (H.S.); chenjie@ahjzu.edu.cn (J.C.); panbanglong@163.com (B.P.)
2   Center of Excellence of Laser Scanning Research, Finnish Geospatial Research Institute, Masala FI-02430, Finland; juha.hyyppa@nls.fi
3   Key Laboratory of Quantitative Remote Sensing Information Technology, Chinese Academy of Sciences, Beijing 100094, China; liwei@aoe.ac.cn (W.L.); hhwu@aoe.ac.cn (H.W.)
4   Interdisciplinary Division of Aeronautical and Aviation Engineering, The Hong Kong Polytechnic University, Kowloon, Hong Kong, China; changhui.jiang1992@gmail.com
\*   Correspondence: yuwei.chen@nls.fi

**Abstract:** Hyperspectral LiDAR (HSL) has been widely discussed in recent years, which attracts increasing attention of the researchers in the field of electronic information technology. With the application of supercontinuum laser source, it is now possible to develop an HSL system, which can collect spectral and spatial information of targets simultaneously. Meanwhile, eye-safety and miniature HSL device with multiple spectral bands are given more priorities in on-site applications. In this paper, we tempt to investigate how to select spectral bands with a selection method. The proposed method consists of three steps: first, the variances among the classes based on hyperspectral feature parameters, termed inter-class variances, are calculated; second, the channels are sorted based on corresponding variances in descending order, and those with the two highest values are adopted as the initial input of classification; finally, the channels are selected successively from the rest of the sorted sequence until the classification accuracy reaches 100%. To test the performance of the proposed method, we collect 91/71-channel hyperspectral measurements of four different categories of materials with 5 nm spectral resolution using an acousto-optic tunable filter (AOTF) based HSL. Experimental results demonstrate that the proposed method could achieve higher classification accuracy than a random band selection method with different classifiers (naïve Bayes (NB) and support vector machine (SVM)) regardless of classification feature parameters (echo maximum and reflectance). To reach 100% accuracy, it demands 8–9 channels on average by echo maximum and 4–5 channels on average by reflectance based on NB classifier; these figures are 3–4 by echo maximum and 2–3 by reflectance with SVM classifier. The proposed method can complete classification task much faster than the random selection method. We further confirm the specific channels for the classification of different materials, and find that the optimal channels vary with different materials. The experimental results prove that the optimal band selection of HSL system for classification is reliable.

**Keywords:** hyperspectral LiDAR; band selection; classification; inter-class variance

## 1. Introduction

Hyperspectral imagery is a passive remote sensing technology, which provides non-invasive and non-intrusive measurement for delicate resolution reflectance characteristics simultaneously with more than a hundred contiguous spectral bands. As image composition materials present their

own corresponding unique spectral profiles in a large number of bands, hyperspectral images mean high-dimensional data [1]. Therefore, one weakness in hyperspectral images is that large amounts of information impose requirements for storage space, computational burdens, and other real-time application problems [2–6]. The second remarkable weakness is that those passive imaging sensors are highly dependent on illumination condition, which limits their extensive applications. The third weakness is their inability to provide information on range or distance, which might be indispensable for some remote sensing applications [7].

As an active sensor, LiDAR is capable to spectral signals from the targets at the specified wavelengths while simultaneously obtaining the ranging information with immunity to illumination changes. Researchers have carried out some investigations of LiDAR for monitoring rice leaf nitrogen, target classification, and other applications [8–10]. So far three types of LiDAR have been applied widely, i.e., monochromatic, multi-spectral and supercontinuum spectral LiDAR. The monochromatic LiDAR obtains the ranging information through measuring the travel time of the laser beam from the laser source to the targets [11]. Such a system would also be necessary for basic studies dealing with the physical effects of backscatter since the coherent backscatter is highly dependent on spectral wavelength [12].

Multi-spectral LiDAR is a combination of several separated lasers operating at different wavelengths and sharing the same optical system. The solution is to combine several monochromatic laser sources of different wavelengths together; but it is hard to address more channels due to the fixed wavelengths. More channels mean more laser sources at different spectral wavelengths; therefore, it is hard to combine tens or hundreds of monochromatic laser sources together [13–16].

A hyperspectral LiDAR (HSL) utilizes a supercontinuum laser source, whose wavelengths cover the visual to short-wavelength infrared bands, for the measurement of distance and reflectance spectrum simultaneously. Able to produce directional broadband light in an optical fiber [17], supercontinuum laser source has been the basis for recent efforts at developing HSL.

The HSL prototype was initially developed for forestry applications in 2007 by Kaasalainen et al. [12]. With the advances in the spectroscopic device technology, Chen and Hakala et al. designed a two-channel hyperspectral LiDAR using optical filters in 2010 [18], and created the first full-waveform HSL system that employed a grating as spectroscopic device to simultaneously generate eight channels with wavelengths ranging from 550 to 1000 nm in 2012 [19]. The acousto-optic tunable filter (AOTF) was capable of filtering with a spectral band from 430 to 1450 nm at a high spectral resolution; therefore, AOTF offered a quicker tuning speed (microseconds) and broader wavelength ranges. Wang et al. designed an 8-channel HSL based on AOTF (AOTF-HSL) with spectrum covering from 450 to 1600 nm [20], and Chen et al. extended channel numbers of HSL to seventeen based on AOTF [21]. Du et al. designed a 32-channel HSL based on a blazed grating with 12 nm resolution for estimation of rice leaf nitrogen contents [13]. Li et al. employed liquid crystal tunable filter (LCTF), which was installed before the avalanche photodiode (APD) for electronically and consecutively selecting the wavelengths of the backscattered echoes, and designed an HSL with 10 nm spectral resolution [7]. A 51-channel AOTF-HSL with spectrally resolved waveform echoes covering 500–1000 nm with 10 nm resolution was designed for the sampling points on the targets [22,23]. Shao and Chen et al. designed a 91-channel AOTF-HSL with 5 nm resolution that spectrum coverage from 650 to 1100 nm for coal/rock classification [24].

The hyperspectral presentation of the targets would provide a large number of spectral features for classification according to point clouds. By increasing the designed channels, HSL improves the accuracy of obtained elevation data, classification, and segmentation of objects, which will make it more flexible and adaptable in the field of application. On the other hand, HSL is accompanied by the problems of data storage and tremendous computational efforts [21,24]. Therefore, if we can design an optimal spectral channels configuration dedicated to a specific application, it will reduce HSL hardware cost, simplify hardware design for a more compact system. Meanwhile, fewer spectral channels would imply better eye safety.

In hyperspectral imagery domain, selecting a relevant range of wavelengths in spectra for classification and some other given tasks is desirable, which can save storage space and simplify image acquisition. Numerous algorithms to minimize bandwidth requirements associated with hyperspectral imaging data have been exploited [8–11]. Scientists have performed various spectral band selections in an unsupervised [25] and a supervised manner [2,26]. Based on the research of hyperspectral imagery band selection, we speculated that the selection of some optimal HSL channels will accelerate computational capabilities and reduce calculation time as well as miniaturizing HSL hardware and meeting eye-safety demands in practical applications.

We have designed an AOTF-HSL for experiments in a controlled laboratory environment, which can acquire proper range accuracy through the whole spectrum. The HSL range is beyond 15 m [24,27], proving that its effective range satisfies practical applications. If the hardware is further simplified, the device will meet the market demand as the selected distance of tens of meters offers a feasible, effective radius for practical survey and records in future with an on-site terrestrial scanning mode. We notice that (1) while designing the HSL, we could not determine which bands are relevant for applications; (2) the acquisition and application of supercontinuum bands require higher energy; and (3) eye-safety is critical in practical applications. Therefore, as lower hyperspectral energy and a miniaturized system are required for on-site cases, we only need to focus on relatively coarse spectral and spatial resolution. If we can design a filter system to select the optimal channels for different applications based on target spectral features, the HSL will be convenient for actual commercial applications. Above all this, we should confirm the feasibility of optimal channel selection to provide a fundamental basis for designing such a filter type. In this paper, we propose an optimal band selection method based on inter-class variances for target classification, aiming to simplify the hardware and acquire high classification accuracy simultaneously. First, we collect hyperspectral records with AOTF-HSL. Then we investigate the proposed method for target classification based on two classifiers: naïve Bayes (NB) and support vector machine (SVM), and test the method with spectral features extracted from hyperspectral measurements.

The rest of this paper includes Sections 2–5. Section 2 introduces the instrument and dataset. Section 3 describes the optimal channel selection method based on inter-class variance. Section 4 presents the detailed experimental results. Section 5 summarizes our research and highlights the open issues for future development.

## 2. Instrument and Dataset Description

### 2.1. AOTF-HSL

In this paper, the experimental instrument employed the AOTF-HSL with 5 nm resolution, which we designed to obtain 91-channel hyperspectral measurements [24]. This research selects a supercontinuum laser source (YSL® SC-OEM) [28] for the lab test. The SC-OEM delivers an ultrawide spectral laser pulse ranging from 400 to 2400 nm with up to 8 W total power at 0.01–1 MHz repetition rate range. The maximum single pulse energy was about 1 μJ, and the full width at half maximum (FWHM) of the transmitted laser pulse was approximately 2 nanoseconds.

The laboratory test employed an YSL® AOTF model as a spectroscopic device. The AOTF module covers the entire supercontinuum spectrum from visible (VIS) to near-infrared (NIR) and shortwave infrared (SWIR) with different filter bandwidth by combining three different crystals. Table 1 presents the specification of the selected AOTF model.

We selected the AOTF spectrum covering from 650 to 1100 nm, and the spectral resolution was 5 nm in this research, which represents a significant improvement in terms of the number of channels available. The time-of-flight (ToF) measurements of different spectral channels were calculated based on the collected waveforms recorded with a simple maximum algorithm [21].

AOTF offers a quicker tuning speed and broader wavelength range, which is preferable for spectral channel selection for a dedicated HSL application. We achieved a continuous spectrum with

a finer spectral resolution of 5 nm and the number of bands in the supercontinuum spectrum will seriously complicate the instrument.

**Table 1.** YSL® acousto-optic tunable filter (AOTF) major parameter specifications.

| Characteristics | Value | | |
|---|---|---|---|
| **AOTF Crystal** | **VIS** | **NIR** | **SWIR** |
| Wavelength Range (nm) | 400–650 | 650–1100 | 1100–2400 |
| Spectral resolution (nm) | 2–7 | 2–6 | 4–16 |
| Response time (μs) | 10 | | |
| AOTF Diffraction Efficiency | >90% | | |
| Output Efficiency | >40% | | |
| Polarization | Linearly Polarized | | |

### 2.2. Dataset Description

We tested four categories of materials with 27 samples, which were measured by the AOTF-HSL with 5 nm resolution. Table 2 lists the related information.

**Table 2.** Dataset description.

| Dataset | Categories | Number of Samples | Number of Channels | Spectrum Coverage (nm) |
|---|---|---|---|---|
| Dataset 1 | coal/rock | 4 | 91 | 650–1100 |
| Dataset 2 | timber | 7 | 91 | 650–1100 |
| Dataset 3 | ore | 6 | 91 | 650–1100 |
| Dataset 4 | plant leaf | 10 | 71 | 650–1000 |

The spectral measurements of samples were collected from the same system, where the spectra of plant leaf experiment in Dataset 4 were 650–1000 nm. We filtered out the signals beyond 1000 nm for their rapid attenuation.

Dataset 1 includes four types of coal/rock samples: coal, rock from roof layer, rock from floor layer, and gangue-rock, which were collected from Wanglou coal mining, Shandong Province, China in 2018 [24].

Dataset 2 consisted of seven species of timber samples: white birch (*Betula platyphylla* Suk.), Chinese ash (*Fraxinus chinensis* Roxb), oak (*Quercus* L.), ailanthus (*Ailanthus altissima* (Mill.) Swingle), goldenrain tree (*Koelreuteria paniculata* Laxm), moso bamboo (*Phyllostachys pubescens* Mazel ex H.de Lehaie), and yunnanensis (*Pinus yunnanensis* Faranch). The timber samples were from the trees cut down after dehydration, and then we collected the hyperspectral measurements on their barks in experiment.

Dataset 3 consisted of six types of ore samples: calcite, fluorite, olivine, orthoclase, plagioclase, and quartz. The samples were the same as Reference [21]; but we tested again with the 91-channel version AOTF-HSL.

Dataset 4 consisted of ten species of plant leaf samples: early lilac (*Syringa oblata* Lindl.), Chinese ash (*Fraxinus chinensis* Roxb), gingko (*Ginkgo biloba* L.), goldenrain tree (*Koelreuteria paniculata* Laxm), Chinese scholar tree (*Sophora japonica* Linn.), honeysuckle (*Lonicera japonica* Thunb.), poplar (*Populus tomentosa* Carr), prunus davidiana (*Amygdalus davidiana* (Carrière) de Vos ex Henry), eucommia (*Eucommia ulmoides* Oliver), and magnolia (*Magnolia denudata* Desr.).

Figure 1 shows some measured samples, namely goldenrain wood (Figure 1a), gingko leaf (Figure 1b), rock from roof layer (Figure 1c), and calcite (Figure 1d).

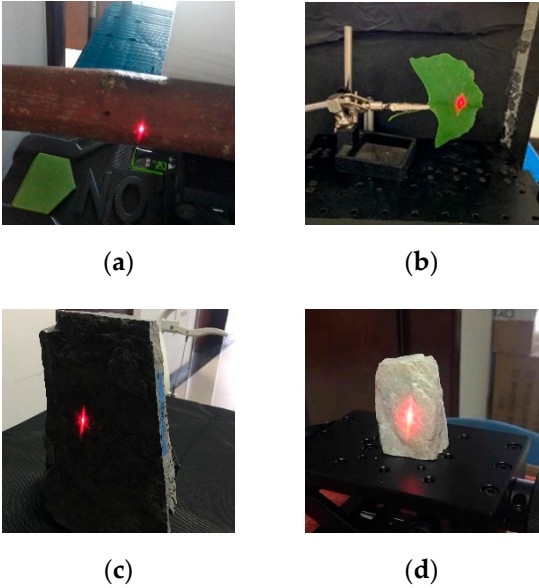

(**a**)  (**b**)

(**c**)  (**d**)

**Figure 1.** Some samples in lab experiment. (**a**) Goldenrain wood; (**b**) gingko leaf; (**c**) rock from roof layer; and (**d**) calcite.

### *2.3. Feature Description*

We employed two features extracted from a spectrally resolved echo waveform, which can record the target reflected powers in a series of continuous channels, namely echo maximum and reflectance. Echo maximum is the amplitude peaks extracted from echo intensity waveforms. LiDAR intensity data have been proven beneficial in data registration, feature extraction, classification, segmentation, and object detection [29]. However, intensity is always influenced by some intrinsic and extrinsic factors. The extrinsic factors of different scanning operations are variable. To overcome this issue, calibration is adopted to correct the recorded intensity values to produce reflectance values that are more useful and more closely related to true surface characteristics. The reflectance is the calibrated echo intensity, which is a prerequisite for many quantitative applications, and it has become an important research topic recently [20–24]. In this paper, we conducted LiDAR calibration assisted with a white reflectance standard board with 70% reflectivity [19,24,27]. Figure 2 shows the spectral profiles of the echo intensity and reflectance maximum from six different samples in Dataset 4.

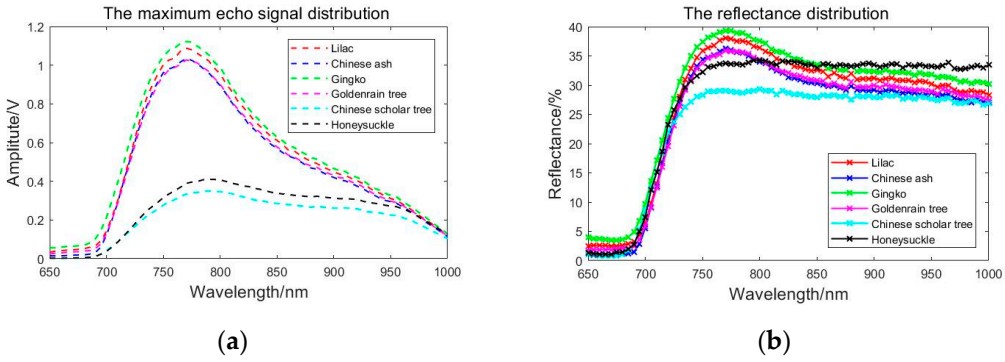

(**a**)  (**b**)

**Figure 2.** The distribution of features. (**a**) Echo maximum and (**b**) reflectance.

### 3. The Proposed Method

To broaden the applications of HSL, we tempted to develop the HSL configuration with a continuous spectrum and high spectral resolution. Classification was conducted based on genuinely hyperspectral data with the excellent results reported in the previous literatures [21,22]. To further

develop the classification performance, we hoped to select several optimal channels to obtain high classification accuracy. To prove the feasibility of spectral band selection, we started from the redundancy of channels.

### 3.1. Redundancy Information

It has been verified that employing the spectral reflectance collected by HSL for targets classification is feasible with abundant spectral information [21]. While we selected reliable classification methods or optimized feature parameters, it is possible that fewer channels will be needed for higher classification accuracy [24]. If we obtain the difference between classes at each channel, prioritize the difference values according to classification results, the optimal channels can be selected. The channel prioritization does not consider the spectral correlation; however, the correlation indicates that hyperspectral measurements contain data redundancy [4,17].

We calculated the cross-correlation coefficient R, which reflects the spectral correlation between each band of two samples. R also denotes the spectral information redundancy of channels [30].

$$R(k) = \frac{1}{(M-k)\sigma(X)\sigma(Y)} \sum_{t=1}^{M-k} (X_t - \mu_X)(Y_{t+k} - \mu_Y). \tag{1}$$

The channels of two samples were selected from $M$ channels of hyperspectral measurements $\mathbf{X_t} \in [x_1, x_2, \ldots x_i, \ldots x_M]$, $\mathbf{Y_t} \in [y_1, y_2, \ldots y_i, \ldots y_M]$, where $x_i = [x_{i1}, x_{i2}, \ldots x_{iN}]$ and $y_i = [y_{i1}, y_{i2}, \ldots y_{iN}]$ are column vectors reshaped by the $i$th band and N is the measurement points of echo signal in temporal domain, $\sigma$ is the standard deviation, $\mu$ is the mean value, $k$ is the difference between the serial number of channels of two samples; and $k$ is less than $M$.

Figure 3 shows the correlation coefficient R of reflectance of two samples (rock from the roof and gangue-rock samples in Dataset 1). The horizontal and vertical coordinates of two spectra were corresponding to each channel; the colors in different depths indicate that values were different. R indicates the correlation between hyperspectral measurements of two samples at different channels. Higher R indicates more redundancy. From the figure, we could conclude that the hyperspectral reflectance values with the 5 nm spectral resolution had redundant values. A rough spectral range was selected based on spatial cross-correlation analysis; however, we still could not determine the contribution rate of each spectral band to the overall spectral characteristics. We could select the major channels to classify while the redundant channels were disued temporarily. The detailed selection process is presented as the following.

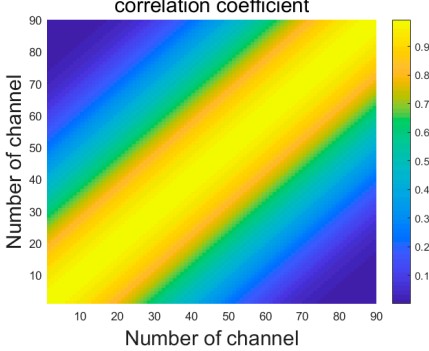

**Figure 3.** Two samples hyperspectral cross-correlation coefficient distribution of 91 channels.

### 3.2. Method Based on Inter-Class Variance

We termed the variance between feature values of two classes as inter-class variance. The high variance means a remarkable difference between these classes, and the classification can obtain the optimal result [31]. We proposed an algorithm to select several optimal bands for classification based

on inter-class variance. Then we selected two feature parameters (echo maximum and reflectance) and employed two classifiers, including NB [32] and SVM [21,24,33].

The inter-class variances for classification were calculated according to the distribution of feature values at each channel. The variances can be regarded as the description of the feature values dispersion degree.

We calculated the inter-class variance in the following equations, which employs the mean of eight measurements rather than a single measurement to mitigate the impact of noise.

$$V_{inter}(j) = \frac{var_j}{var_{max}} \quad , $$
$$var_j = \left[ \frac{1}{L} \sum_{i=1}^{L} \left( value_{i,j} - \overline{value_{i,j}} \right)^2 \right]^{\frac{1}{2}} \tag{2}$$

where *value* refers to echo maximum, reflectance, and other feature parameters; *j* denotes the serial number of the current channel, $j \in [1, 2, \ldots, M]$, M is the maximum number of HSL channels; $i \in [1, 2, \ldots, L]$, and L represents the total number of classes.

The variances at all wavelengths were calculated based on hyperspectral measurements, which were normalized by maximum variance to obtain $V_{inter}$. The channels were sorted based on $V_{inter}$ in descending order in the channel sets for classification, and those with the two highest values were adopted as the initial input. The selected channel set grew until the classification accuracy reach 100%; the selected channels in the set were the optimum selected channels. We will present the optimum spectral band selection procedure for classification base on the $V_{inter}$ values in the next step.

In a nutshell, the whole flowchart of the proposed Method (Matlab style) was as follows.

---

**Method 1.** Pseudo Code of the Proposed Method

---

**Input**: Set the indicating variable m=M (which is 91 or 71 in this study), initialize alternative index set as the full channels, i.e. alternative index set *idx* = {1, 2, 3, . . . ., m}, and initialize the selected channels index set *index* = {ø}.

---

% Main loop
while the stop condition is not met
**Step 1**: Calculate $V_{inter}(idx)$ among the classes of all the channels, using Equation (2).
**Step 2:** Find maximum and second highest values of $V_{inter}$ ($n_j$) in $V_{inter}(idx)$, send the index of the channel to initial index set *initial* = {$n_i$, $n_j$}, and *index* = {$n_i$, $n_j$}. The alternative set of channel index *idx* = {1, 2, 3, . . . , $n_{i-1}$, $n_{i+1}$, $n_{j-1}$, $n_{j+1}$, . . . , . . . , m}.
**Step 3:** Find maximum of $V_{inter}$ ($n_d$) in $V_{inter}(idx)$, send $n_d$ to the selected index set, *index* = {$n_d$, *index*}.
**Step 4:** Remove $n_d$ form the alternative set, the new alternative set *idx* = {*idx*}_{$n_d$}.
**Step 5:** Calculate the result of multiple classification with features corresponding to *initial* channels.
　if the result reaches 100%,
　　Output the selected set {$n_i$, $n_j$}
　else
　　go to **Step 6**
　end while
**Step 6:** Calculate the result of multiple classification with features corresponding to *index* channels.
　if the result reach 100%,
　　Output the selected set {$n_d$, $n_i$, $n_j$}
　else
　　go to **Step 3**
　end while
Output the optimal channels corresponding to the selected set by maximum, sequenced as
*index* = {$n_1$, $n_2$, $n_j$, . . . , $n_q$}.
**Step 7**: Sort values of $V_{inter}$ (*index*), send the channel index to the optimal set in turn,
*index2* = {$n_{m1}$, $n_{m2}$, . . . ., $n_{mq}$}.
**Output:** The optimal channels corresponding to the optimal set, sequenced as *index2* = {$n_{m1}$, $n_{m2}$, . . . ., $n_{mq}$}.

---

## 4. Results and Analysis

### 4.1. Classification Performance

　　Two parameters of echo maximum and reflectance were utilized to evaluate the performance of the proposed method for classification. Table 3 lists classification results with two channel selection methods based on echo maximum as feature parameter, where the figures in the table indicate the minimum number of spectral channels (MNSC) used to acquire 100% classification accuracy [24]. Channels were selected randomly to classify the samples, and we termed the method as Normal Method. All classifiers can achieve excellent accuracy with echo maximum directly while abundant spectral information is available. The MNSC of Dataset 1, Dataset 2, Dataset 3, and Dataset 4 based on the normal method was 47, 37, 55, and 33 respectively with NB, and 12, 6, 16, and 24 with SVM respectively.

**Table 3.** Minimum number of spectral channels (MNSC) comparison of classification with echo maximum.

| Dataset | NB | | SVM | |
|---|---|---|---|---|
| | **Normal Method** | **The Proposed Method** | **Normal Method** | **The Proposed Method** |
| Dataset 1 | 47 | 9 | 12 | 4 |
| Dataset 2 | 37 | 10 | 6 | 3 |
| Dataset 3 | 55 | 2 | 16 | 3 |
| Dataset 4 | 33 | 13 | 24 | 5 |
| average | **43** | **8.5** | **14.5** | **3.75** |

　　The MNSC of Dataset 1, Dataset 2, Dataset 3, and Dataset 4 based on the proposed method was 9, 10, 2, and 13 respectively with NB and 4, 3, 3, and 5 respectively with SVM. It is worth noticing that the proposed method outperformed the normal method. With NB classifier, the MNSC of the proposed method was 38, 27, 53, the 20 less than that of the normal method, reduced by 80.85%, 72.97%, 96.36%, and 60.6% respectively. The similar promising results could also be found that the MNSC dropped to 8, 3, 13, and 19, reduced by 66.67%, 50%, 81.25%, 79.17% respectively with SVM. On average, only 8–9 channels with NB and 3–4 channels with SVM reached 100% classification accuracy.

　　Table 4 lists classification results with reflectance as the feature parameter. The MNSC of Dataset 1, Dataset 2, Dataset 3, and Dataset 4 based on the normal method was 30, 7, 24, and 19 respectively with NB, and 7, 5, 7, and 22 respectively with SVM.

**Table 4.** MNSC comparison of classification with reflectance.

| Dataset | NB | | SVM | |
|---|---|---|---|---|
| | **Normal Method** | **The Proposed Method** | **Normal Method** | **The Proposed Method** |
| Dataset 1 | 30 | 6 | 7 | 3 |
| Dataset 2 | 7 | 3 | 5 | 2 |
| Dataset 3 | 24 | 2 | 7 | 2 |
| Dataset 4 | 19 | 7 | 22 | 3 |
| Average | **20** | **4.5** | **10.25** | **2.5** |

　　The MNSC of Dataset 1, Dataset 2, Dataset 3, and Dataset 4 based on the proposed method was 6, 3, 2, and 7 with NB, and 3, 2, 2, and 3 with SVM. As can be seen, there were fewer channels needed based on the proposed method regardless of NB and SVM classifiers. The MNSC based on the proposed method was 24, 4, 22, and 12 less than the normal method with NB, reduced by 80%, 57.14%, 91.66%, and 63.15% respectively. The MNSC dropped to 4, 3, 4, and 19, reduced by 57.14%, 60%, 71.43%, and 86.37% respectively with SVM. On average, only 4–5 channels were needed with NB and 2–3 channels with SVM to reach 100% classification accuracy.

　　The proposed method could select several optimal spectral channels for target classification. Several optimal channels selected from 91/71 channels to classify the targets will speed up the

classification and potentially simplify the practical HSL hardware design. While the new category samples are added, the algorithm will remain valid to determine the optimal channels by calculating inter-class variances.

### 4.2. The Selected Channels

The accuracy of the standard classifiers could reach 100% when three or four channels are selected using the proposed method. Table 5 lists the selected channel using the proposed method with NB and SVM classifiers, including two feature parameters. If $n$ denotes serial number of a channel, the wavelength of the channel could be calculated by $650 + 5(n − 1)$ nm.

**Table 5.** The comparison of selected channels.

| Dataset | Echo Maximum | Reflectance |
|---------|--------------|-------------|
| Dataset 1 | {**5, 12, 16, 21,** 33, 46, 52, 59, 75} | {**21, 5, 12, 16,** 33} |
| Dataset 2 | {**12, 15, 22,** 38, 41, 45, 49, 53, 59, 67} | {**7, 11,** 6} |
| Dataset 3 | {**3, 7**} | {**2, 1**} |
| Dataset 4 | {**17, 16, 18, 19, 15,** 20, 2, 14, 13, 21, 62, 67, 1} | {**2, 17, 16,** 3, 1, 4, 5} |

All figures in the braces {·} indicate the number of selected channels with NB classifier, and the boldfaced blue ones represent the selected channels with SVM. We could see that the input feature parameters affected selected results directly with reflectance leading to better results than echo maximum.

Figure 4 presents the selected spectral bands of different datasets with echo maximum. In on-site applications, it is inconvenient to obtain the reference board, thus we selected the information extracted from echo intensity signals with NB classifier, which is a straightforward method for supervised learning.

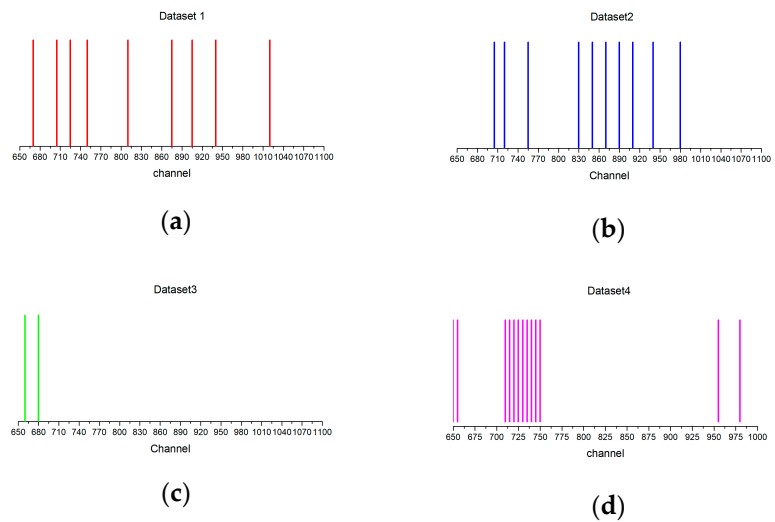

**Figure 4.** The selected bands with echo maximum by naïve Bayes (NB) classifier. (**a**) Dataset 1; (**b**) Dataset 2; (**c**) Dataset 3; and (**d**) Dataset 4.

We found that the selected channels for different materials varied. For the coal/rock samples in Dataset 1, the selected spectrum was from 670 to 1020 nm, and the minimum interval was 20 nm, which means an 18-band miniaturized LiDAR with 20 nm resolution should suffice. For the timber samples in Dataset 2, the spectrum was from 705 to 980 nm, and the minimum interval was 15 nm; thus a 19-channel miniaturized LiDAR with 15 nm resolution would meet classification requirements. The spectra for ore samples classification cover from 650 to 680 nm with six channels with 5 nm resolution.

The 655–740 nm range for plant leaves in Dataset 4 was consistent with the previous studies in that the red edge position usually locates in red band spectrum normally from 680 to 720 nm [13,23,34,35]. Chlorophyll contained in plants absorbs a large amount of light in the visible spectrum and it becomes almost transparent at wavelengths greater than 700 nm [13]. The results imply that the design of future measurement equipment could be planned.

To evaluate the processing speed of the proposed method, we compared the computing time for classification of two methods. We took the timber samples in Dataset 2 based on reflectance as an example, which had the smallest MNSC; therefore, the calculation time was minimal. The MNSC of the normal method was 5, and the classification time was 0.6043 s. The method starts with 2 channels, then 3, and 4 channels to test whether results can reach 100%. The classification time was 10.2436 s, 5.755 s, and 2.0787 s with averaging 20 experiments, respectively. Therefore, the time totaled 18.6816 s by the normal method. By contrast, the calculating time was around 0.83 s when we employed the proposed method. We thus concluded that the proposed method could significantly reduce the processing time.

## 5. Conclusions

Our investigation was conducted to select the optimal channels that typically provide 100% classification accuracy with rapid data processing based on the proposed method under two hyperspectral feature constraints. While selecting the crucial and significant channels to classify the targets, the complexity, computing time, and hardware investment of HSL would considerably decrease. Our proposed method clearly demonstrated that hyperspectral LiDAR systems with fewer optimal channels could obtain the desired results in target classification applications.

Our future work is to design a miniaturized version of AOTF-HSL with a band selection filter based on applications. Based on hardware designed, we will discuss the relations and differences of spectral features of heterogeneous materials. Additionally, to simplify the test, the laser source is perpendicular to the target in this experiment, while in the future we will test the HSL with the laser source pointing to the targets from different directions. We will also improve the bands from 950 to 1100 nm to verify its generic applicability with current configuration.

**Author Contributions:** Conceptualization and writing original draft preparation: H.S. and Y.C.; methodology, supervision, project administration, funding acquisition, J.H., Y.C., J.C.; field test: W.L., H.W., data analysis: W.L., C.J. and B.P., writing—review and editing supported by J.H. and J.C. All authors have read and agreed to the published version of the manuscript.

**Funding:** The author gratefully acknowledges the financial support from Academy of Finland projects "Centre of Excellence in Laser Scanning Research (CoE-LaSR; 307362)", "New laser and spectral field methods for in situ mining and raw material investigations (project 292648), (projects 314177 and 307929). This work was supported in part by Anhui Natural Science Research Foundation (KJ2019A0767, 1804d08020314, KJ2017A500, 1708085MD90), Beijing Municipal Science and Technology Commission (Z181100001018036), Research Program of Anhui Jianzhu University (JZ192007), Anhui and Jiangsu Province Key Laboratory Research Found (2017kfkt009, 2019-157).

**Conflicts of Interest:** The authors declare no conflict of interest.

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
