# Peer review of "An Investigation of Spectral Band Selection for Hyperspectral LiDAR Technique"

_electronics, doi:10.3390/electronics9010148_

Round 1

Reviewer 1 Report

Dear authors,

By readding your manuscript, I was more looking for an algorithm to provide the best bands on board, because the application-based results might be different depending on the homogeneity/heterogeneity of the study area. I have listed some of my comments below:

Line 17: “and other applications” -> “and related applications”

Line 20 :” we are more concern about” -> “we are more concerned about”

Line 21-22 :  “feasibility of application based” -> “feasibility of such system based”

Line 22: “In the paper,” -> “In this paper,”

Line 23: “for classification purpose” -> “for classification purposes”

Line 27: “the channels selected successively” -> “the channels were selected successively”

Line 29: “categories materials” -> “categories of materials”

Line 35: “on average reach” -> “on average reached”

Line 36: “with NB classifier. While” -> “with the NB classifier, while”

Line 37: “feature with SVM” -> “feature using the SVM”

Line 38-39: “the proposed method can complete classification task much faster than traditional method” -> what do you mean by “traditional method”?

Line 39: “classification different” -> “classification of different”

Line 39-40: “find the different materials” -> “find that for different materials”

Line 40: “The experiment results” -> “The experimental results”

Line 46: “remote sensing technique”-> “remote sensing technology”

Line 50: “high-dimensionality data”-> “high-dimensional data”

Line 60: “nitrogen and classifying targets and” ->” nitrogen, target classification and”

Line 63: “measuring time travelling by the”-> “measuring the travel time of the”

Line 65: “coherent backscatter should be highly”-> “coherent backscatter is highly”

Line 67: “wavelengths which is a combination” -> “wavelengths is a combination”

Line 71: “spectral wavelength”-> “spectral wavelengths”

Line 79-80: “the HSL prototype has been initially” -> “the HSL prototype was initially”

Line 80: “applications firstly in 2007”-> “applications in 2007”

Line 83: “HSL system employing a” -> “through employing a OR that employed a”

Line 84: “with wavelength ranging (550 nm-1000 nm)” -> “with the wavelengths ranging from 550 nm-1000 nm

Line 96: “designed a 91-channel 5 nm resolution AOTF-HSL with continuous spectrum coverage from” REWRITE THIS

Line 99:  “By increasing the receiving channels”->” By increasing the designed channels”

Line 100:  “obtained elevation models”-> “obtained elevation data”. Laser scanners do not provide elevation model at first and these models are a result of processing their elevation data.

Line 106: “selecting the relevant range”-> “selecting relevant range”

Line 108-111: “Numerous algorithms to minimize bandwidth requirements associated with hyperspectral imaging data have exploited [8–11], two categories of hyperspectral band selection methods are generally used: unsupervised methods [26] and supervised methods [2, 27].” it is better to split this into two sentences

Line 113: “time, also will miniaturize the” -> “time, as well as miniaturizing the”

Line 121: “which channels is important” -> “which channels are important”

Line 122: “practical application” -> “practical applications”

Line 125: “different application” -> “different applications”

Line 126: “commercial application” -> “commercial applications”

Line 127: “such type filter”-> “such a filter type”

Line 127: “In the” -> “In this”

Line 130: “HSL application”-> “HSL applications”

Line 130: “the fields required with” -> “the fields that require”

Line 131-132: “and acquire the accuracy of the”. What do you mean?

Line 132: “In the paper” -> “In this paper”

Line 133: “resolution AOTF-HSL firstly.” UNCLEAR

Line 134: “for targets classification” -> “for target classification”

Line 136: “of different categories samples” -> “of samples from different categories”

Line 143: “In the paper” -> “In this paper”

Line 154 (table 1): what are the values for vis & swir?

Line 157: “was set on 5 nm”-> “was set to 5 nm

Line 157: “significant advance” -> “significant improvement”

Line 168: “of the AOTF-HSL” -> “of AOTF-HSL”

Line 168-169: “algorithm in the next section, we” -> “algorithm, we”

Line 169: “categories materials”-> “categories of materials”

Line 169: “samples as four datasets”-> “samples”

Line 170:  “resolution, the related information listed in Table”  -> “resolution. The related information is listed in Table”

Line 175: “samples collected” -> “samples were collected”

Line 175-177: “AOTF-HSL, where the spectrum of plant leaf experiment was 650-1000 nm, for the signals were week beyond 1000 nm, so we filtered them in test.” -> UNCLEAR

Line 179: “kinds coal” -> “kinds  of coal”

Line 186-187: “The trees cut down as timber after suitable dehydration for next application, we test whether classify them form bark appearance.” UNCLEAR

Line 188: “Dataset 3 is six” -> “Dataset 3 consists of six”

Line 189: “The samples same to”-> “The samples are same to”

Figure 1: figures are not aligned

Line 203: “Where maximum” -> “Maximum”

Line 204-206: what do you mean? using reflectance has been a hot topic recently?

Line 206: “In the paper” -> “In this paper”

Line 206-207: “paper, LiDAR calibration conducted assisted with a”. What do you mean?

Figure 2: how can you describe this figure? what is the relationship between amplitude and reflectance?

Line 216: “channel select” -> “channel selection”

Line 221: “While we select the excellent classifiers” -> there is no excellent classifier?

use Formal  words, like common and remove “the”

line 226: “information is the basis of keeping significant information for” REWRITE THIS

line 231: “channels hyperspectral” -> “channels of hyperspectral”

line 233: “k is the number difference between the” what do you mean by the number of differences?

Line 237: “by the maximum values to” -> maximum values of WHAT?

Line 240: “channels, the higher”-> “channels, in which the higher”

Line 249: “difference in classes” -> “difference between these classes”

In EQ 2; what is valueI,J here? define these variables?

Line 262: “and Vinter values are” what is Vinter? you must define all variables in the text

Line 264-265: “input, then second-largest variance value is included in the set, and so on.” Varinace is defined between two classes. So, how do you sort them? They might have correlations and these number are not probably independent!

Line 266: “selected channels selected” remove selected

In pseudo code: “use equation (2).”-> “using equation (2).”

Remove the first end while in Step 4 of the pseudo code

In Step 4: “to the select set”-> “to the selected set”

In Step 5: “envelope of Vinter”-> “envelope for Vinter”

In Step 5: “to the select set”-> “to the selected set”

In Output: “the select set”-> “the selected set”

Line 272: “To honestly evaluate the proposed algorithm performance for” -> “To evaluate the performance of the proposed algorithm for”

Line 274: “results with different channels select method based” what do you mean?

Line 280: “results, which” -> “results, in which”

In Table 3; what do you mean by these flout values for number of bands?

Line 292: “that the results increase based on the proposed selected algorithm” you mean the number of bands?

Line 295:” dropped to 8,”-> “dropped by 8,”

Line 296: “channels reach”-> “channels that reach”

Line 299: “Table 4 lists classification results with”-> there is no classification result in this table? only the minimum number of bands are reported here

Line 305: “the results considerably increased based on” what is increased? what results?

Line 308: “In average,” -> “On average,”

Line 308: “channels reach”-> “channels that reach”

Line 309: “channels reach 100%” -> “channels that reach 100%”

Line 309-310: “Nevertheless, it is further confirmed that spectral channels which be suited to target classification.” check this sentence. I cannot imagine what do you want to say!

Line 315: “classification is suitable” -> “classification are suitable”

Line 318-319: “Besides, the exact bands could be provided, we listed them as following.” REWRITE

Line 321: “channels, and the wavelength” -> “channels, the wavelength”

Line 325: “SVM classifier”-> “SVM classifier as well”

Line 326-327: “input feature parameters affect selected results directly, reflectance as features obtain better results, for the reflectance is optimized parameter based on echo signal.” Unclear

Line 330: “extract from”-> “extracted from”

Line 333: “channels with different”-> “channels for different”

Line 336: ‘The timber samples in Dataset 2 is from’->“For the timber samples in Dataset 2, it is from”

Line 338-339: “The ore samples classification spectral range from 650 nm to 680 nm with 6 channels with 5 nm resolution.” what is the verb of this sentence?

Line 346: “The computing time for the classification”-> “The computing time for classification”

Line 348: “, for the calculated time was least.” What do you mean?

Line 351: “, for they cannot reach 100% accuracy”-> what do you mean?

Line 354: “seconds employed”-> “seconds when employed”

Line 359-360: “, however, the channels selected applications to hyperspectral LiDAR exploitation has not yet to be explored” what do you mean?

Line 362: “channels typically”-> “channels that typically”

Line 369: “technique for hyperspectral LiDAR” so do you mean this can be as a postprocessing step? or on-board?

Line 371:  “LiDAR can be simplified completely determined based” What do you mean?

Line 372: “version AOTF-HSL for channels select filter”-> “version of AOTF-HSL for channels select filter” and what do you by the second part (channels select filter)

Line 375 and 376: change “form” to “from”

Author Response

Line 17: “and other applications” -> “and related applications”

Reply: Fixed. Thank you.

Line 20 :” we are more concern about” -> “we are more concerned about”

Reply: Fixed.

Line 21-22 : “feasibility of application based” -> “feasibility of such system based”

Reply: Fixed.

Line 22: “In the paper,” -> “In this paper,”

Reply: Fixed.

Line 23: “for classification purpose” -> “for classification purposes”

Reply: Fixed.

Line 27: “the channels selected successively” -> “the channels were selected successively”

Reply: Fixed.

Line 29: “categories materials” -> “categories of materials”

Reply: Fixed.

Line 35: “on average reach” -> “on average reached”

Reply: Fixed.

Line 36: “with NB classifier. While” -> “with the NB classifier, while”

Reply: Fixed.

Line 37: “feature with SVM” -> “feature using the SVM”

Reply: Fixed.

Line 38-39: “the proposed method can complete classification task much faster than traditional method” -> what do you mean by “traditional method”?

Reply: “Traditional method” is inexact, we modified it as randomly selected channels method, which we termed it as Normal Method in the section 4. Thank you for mentioning.

Line 39: “classification different” -> “classification of different”

Reply: Fixed.

Line 39-40: “find the different materials” -> “find that for different materials”

Reply: Fixed.

Line 40: “The experiment results” -> “The experimental results”

Reply: Fixed.

Line 46: “remote sensing technique”-> “remote sensing technology”

Reply: Fixed.

Line 50: “high-dimensionality data”-> “high-dimensional data”

Reply: Fixed.

Line 60: “nitrogen and classifying targets and” ->” nitrogen, target classification and”

Reply: Fixed.

Line 63: “measuring time travelling by the”-> “measuring the travel time of the”

Reply: Fixed.

Line 65: “coherent backscatter should be highly”-> “coherent backscatter is highly”

Reply: Fixed.

Line 67: “wavelengths which is a combination” -> “wavelengths is a combination”

Reply: Fixed.

Line 71: “spectral wavelength”-> “spectral wavelengths”

Reply: Fixed.

Line 79-80: “the HSL prototype has been initially” -> “the HSL prototype was initially”

Reply: Fixed.

Line 80: “applications firstly in 2007”-> “applications in 2007”

Reply: Fixed.

Line 83: “HSL system employing a” -> “through employing a OR that employed a”

Reply: Fixed.

Line 84: “with wavelength ranging (550 nm-1000 nm)” -> “with the wavelengths ranging from 550 nm-1000 nm”

Reply: Fixed.

Line 96: “designed a 91-channel 5 nm resolution AOTF-HSL with continuous spectrum coverage from” REWRITE THIS

Reply: We rewrote the sentence, “we designed a 91-channel AOTF-HSL with 5 nm resolution that spectrum coverage from 650 nm to 1100 nm for coal/rock classification ”, Thank you.

Line 99: “By increasing the receiving channels”->” By increasing the designed channels”

Reply: Fixed.

Line 100: “obtained elevation models”-> “obtained elevation data”. Laser scanners do not provide elevation model at first and these models are a result of processing their elevation data.

Reply: Fixed. Thank you.

Line 106: “selecting the relevant range”-> “selecting relevant range”

Reply: Fixed.

Line 108-111: “Numerous algorithms to minimize bandwidth requirements associated with hyperspectral imaging data have exploited [8–11], two categories of hyperspectral band selection methods are generally used: unsupervised methods [26] and supervised methods [2, 27].” it is better to split this into two sentences

Reply: Thank you for suggestion. We rewrote this sentence.

Line 113: “time, also will miniaturize the” -> “time, as well as miniaturizing the”

Reply: Fixed.

Line 121: “which channels is important” -> “which channels are important”

Reply: Fixed.

Line 122: “practical application” -> “practical applications”

Reply: Fixed.

Line 125: “different application” -> “different applications”

Reply: Fixed.

Line 126: “commercial application” -> “commercial applications”

Reply: Fixed.

Line 127: “such type filter”-> “such a filter type”

Reply: Thank you. Fixed.

Line 127: “In the” -> “In this”

Reply: Fixed.

Line 130: “HSL application”-> “HSL applications”

Reply: Fixed.

Line 130: “the fields required with” -> “the fields that require”

Reply: Fixed.

Line 131-132: “and acquire the accuracy of the”. What do you mean?

Reply: We modified them as “high classification accuracy”.

Line 132: “In the paper” -> “In this paper”

Reply: Fixed.

Line 133: “resolution AOTF-HSL firstly.” UNCLEAR

Reply: We rewrote the sentence, thank you for suggestion.

Line 134: “for targets classification” -> “for target classification”

Reply: Fixed.

Line 136: “of different categories samples” -> “of samples from different categories”

Reply: Fixed.

Line 143: “In the paper” -> “In this paper”

Reply: Fixed.

Line 154 (table 1): what are the values for vis & swir?

Reply: The Wavelength Range of VIS and SWIR is listed in Table 1.

Line 157: “was set on 5 nm”-> “was set to 5 nm”

Reply: Fixed.

Line 157: “significant advance” -> “significant improvement”

Reply: Fixed.

Line 168: “of the AOTF-HSL” -> “of AOTF-HSL”

Reply: Fixed.

Line 168-169: “algorithm in the next section, we” -> “algorithm, we”

Reply: Fixed.

Line 169: “categories materials”-> “categories of materials”

Reply: Fixed.

Line 169: “samples as four datasets”-> “samples”

Reply: Fixed.

Line 170: “resolution, the related information listed in Table” -> “resolution. The related information is listed in Table”.

Reply: Fixed.

Line 175: “samples collected” -> “samples were collected”

Reply: Fixed.

Line 175-177: “AOTF-HSL, where the spectrum of plant leaf experiment was 650-1000 nm, for the signals were week beyond 1000 nm, so we filtered them in test.” -> UNCLEAR

Reply: We rewrote the sentence. The spectra of plant leaf experiment in Dataset 4 were 650-1000 nm, we filtered out the signals beyond 1000 nm for their rapid attenuation.

Line 179: “kinds coal” -> “kinds of coal”

Reply: Fixed.

Line 186-187: “The trees cut down as timber after suitable dehydration for next application, we test whether classify them form bark appearance.” UNCLEAR

Reply: We rewrote the sentence as “The timber samples were the trees cut down after suitable dehydration, we collected the hyperspectral measurement form their barks in experiment.”

Line 188: “Dataset 3 is six” -> “Dataset 3 consists of six”

Reply: Fixed.

Line 189: “The samples same to”-> “The samples are same to”

Reply: Fixed.

Figure 1: figures are not aligned

Reply: We modified the figures, thank you.

Line 203: “Where maximum” -> “Maximum”

Reply: Fixed.

Line 204-206: what do you mean? using reflectance has been a hot topic recently?

Reply: LiDAR intensity is always influenced by some intrinsic and extrinsic parameters, more specifically, the intrinsic parameters include power of the emitted laser beam, atmospheric attenuation, and the extrinsic parameters contain reflectivity of the target, transmitting range and incident angle. So, the reference was utilized to evaluate the accuracy of the spectral profile measured by the HSL. The intensity is converted into reflectance by applying the distance and spectral calibration. During our calibration measurements, waveforms are collected using a 70% Spectralon© white reflectance board as a reference target at various distances. The echo intensities are normalized with the intensity of the Spectralon echo at the same distance, producing reflectance.

Line 206: “In the paper” -> “In this paper”

Reply: Fixed.

Line 206-207: “paper, LiDAR calibration conducted assisted with a”. What do you mean?

Reply: Due to the complexity of the AOTF-HSL optical design, the authentic range cannot be easily measured with the current setup. The range precision or stability was evaluated instead of range accuracy by measuring the identical range of standard write board.

LiDAR calibration can be achieved by passing a sample of the transmitted signal through the receiver and monitoring the signal from the standard targets, due to the Lambertian reflectance characteristics and known reflectance values. In this paper, the intensity of the AOTF-HSL is calibrated in the laboratory with a Spectralon© white reflectance standard board.

Figure 2: how can you describe this figure? what is the relationship between amplitude and reflectance?

Reply: Figure 2 shows reflectance and echo intensities, which are selected as feature parameters in target classification. The echo intensities are normalized with the intensity of the reference echo at the same distance, producing reflectance. During the calibration measurements, waveforms are collected using a 70% Spectralon© white reflectance standard board as a reference target at various distances.

LiDAR intensity data have proven beneficial in data registration, feature extraction, classification, surface analysis, segmentation, and object detection and recognition. However, LiDAR intensities the extrinsic factors including the laser transmitting range, laser incident angle and material reflectivity are variable for different scanning operations. To overcome this issue, the calibration are adopted to correct the recorded intensity values to produce values that are more useful and more closely related to true surface characteristics (reflectance).

Line 216: “channel select” -> “channel selection”

Reply: Fixed.

Line 221: “While we select the excellent classifiers” -> there is no excellent classifier?use Formal words, like common and remove “the”

Reply: Thank you for advice. We modified the sentence.

line 226: “information is the basis of keeping significant information for” REWRITE THIS

Reply: Thank you for advice. We rewrote the sentence.

line 231: “channels hyperspectral” -> “channels of hyperspectral”

Reply: Fixed.

line 233: “k is the number difference between the” what do you mean by the number of differences?

Reply: The channel number is 91/71, Suppose the channel number of the sample one is n1={1, 2,...91} and another sample is n2={1, 2,...91} ,and the number difference of the two samples is k =n1-n2. We modified the paragraph.

Line 237: “by the maximum values to” -> maximum values of WHAT?

Reply: Fixed.

Line 240: “channels, the higher”-> “channels, in which the higher”

Reply: Fixed.

Line 249: “difference in classes” -> “difference between these classes”

Reply: Fixed.

In EQ 2; what is valueI,J here? define these variables?

Reply: We rewrote the EQ(2) by replacing Vj(j) with Vinter(j). Vinter(j) is the normalized variance between two classes. Thank you for mentioning.

Line 262: “and Vinter values are” what is Vinter? you must define all variables in the text

Reply: Thank you. We defined the Vinter in the EQ(2).

Line 264-265: “input, then second-largest variance value is included in the set, and so on.” Varinace is defined between two classes. So, how do you sort them? They might have correlations and these number are not probably independent!

Reply: The variances at all wavelengths are calculated based on hyperspectral measurements (reflectance and echo intensities) among all classes. In this paper, we test four categories of materials, and the number of classes is 4, 7, 6, 10 respectively. We calculate variances of hyperspectral measurement values at each channel, and then sort them.

Line 266: “selected channels selected” remove selected

Reply: Fixed.

In pseudo code: “use equation (2).”-> “using equation (2).”

Reply: Fixed.

Remove the first end while in Step 4 of the pseudo code

Reply: Fixed.

In Step 4: “to the select set”-> “to the selected set”

Reply: Fixed.

In Step 5: “envelope of Vinter”-> “envelope for Vinter”

Reply: Fixed.

In Step 5: “to the select set”-> “to the selected set”

Reply: Fixed.

In Output: “the select set”-> “the selected set”

Reply: Fixed.

Line 272: “To honestly evaluate the proposed algorithm performance for” -> “To evaluate the performance of the proposed algorithm for”

Reply: Fixed.

Line 274: “results with different channels select method based” what do you mean?

Reply: Table 3 lists classification results with two channel selection method (Normal method and the proposed method).

Line 280: “results, which” -> “results, in which”

Reply: Fixed.

In Table 3; what do you mean by these flout values for number of bands?

Reply: The proposed algorithm input the number of channels based on 100% classification accuracy. We classify the materials with 2 channels firstly, then 3 channels and so on, until the classification results reach 100%, output the number of channels, as the number in the Table 3.

Line 292: “that the results increase based on the proposed selected algorithm” you mean the number of bands?

Reply: The proposed algorithm only need 8-9 channels that reach 100% accuracy with NB and only 3-4 channels that reach 100% accuracy with SVM. The results outperform Normal Method. To describe accurately, we rewrote the sentence.

Line 295:” dropped to 8,”-> “dropped by 8,”

Reply: Fixed.

Line 296: “channels reach”-> “channels that reach”

Reply: Fixed.

Line 299: “Table 4 lists classification results with”-> there is no classification result in this table? only the minimum number of bands are reported here

Reply: Table 4 lists classification results is the minimum number of spectral channels (MNSC) as classification results. The number is minimum number of channels used to achieve 100% classification accuracy, and it is basis of the Hyperspectral LiDAR channels selection.

Line 305: “the results considerably increased based on” what is increased? what results?

Reply: The results show that fewer channels needed to reach 100% accuracy the based on the proposed algorithm. We rewrote the sentence.

Line 308: “In average,” -> “On average,”

Reply: Fixed.

Line 308: “channels reach”-> “channels that reach”

Reply: Fixed.

Line 309: “channels reach 100%” -> “channels that reach 100%”

Reply: Fixed.

Line 309-310: “Nevertheless, it is further confirmed that spectral channels which be suited to target classification.” check this sentence. I cannot imagine what do you want to say!

Reply: The sentence means “Several optimal spectral channels selected for target classification by using the proposed algorithm.” and we described in detail in next paragraph, so we deleted it.

Line 315: “classification is suitable” -> “classification are suitable”

Reply: Fixed.

Line 318-319: “Besides, the exact bands could be provided, we listed them as following.” REWRITE

Reply: We rewrote the sentence, thank you for advice.

Line 321: “channels, and the wavelength” -> “channels, the wavelength”

Reply: Fixed.

Line 325: “SVM classifier”-> “SVM classifier as well”

Reply: Fixed.

Line 326-327: “input feature parameters affect selected results directly, reflectance as features obtain better results, for the reflectance is optimized parameter based on echo signal.” Unclear

Reply: We modified the sentence. Thank you for mentioning.

Line 330: “extract from”-> “extracted from”

Reply: Fixed.

Line 333: “channels with different”-> “channels for different”

Reply: Fixed.

Line 336: ‘The timber samples in Dataset 2 is from’->“For the timber samples in Dataset 2, it is from”

Reply: Fixed.

Line 338-339: “The ore samples classification spectral range from 650 nm to 680 nm with 6 channels with 5 nm resolution.” what is the verb of this sentence?

Reply: Fixed.Thank you for advice.

Line 346: “The computing time for the classification”-> “The computing time for classification”

Reply: Fixed.

Line 348: “, for the calculated time was least.” What do you mean?

Reply: As listed in Table 4, the MNSC of the dataset 2 was smallest, so the calculated time was least.

Line 351: “, for they cannot reach 100% accuracy”-> what do you mean?

Reply: We rewrote the sentence “for the training process needed more time when they cannot reach 100% accuracy.” In test, we found classification with 100% accuracy need fewer time than classification accuracy less than 100%.

Line 354: “seconds employed”-> “seconds when employed”

Reply: Fixed.

Line 359-360: “, however, the selected channelsapplications to hyperspectral LiDAR exploitation has not yet to be explored” what do you mean?

Reply: With the increase of the designed channels, hyperspectral LiDAR improves the accuracy of obtained elevation data, classification and segmentation of objects, which will make it more flexible and adaptable in the field of application. In previous studies the researchers concerned to obtain consecutive and high-spectral-resolution laser pulses in time dimension and more spectral channels which is the base of next application. The selection of some important hyperspectral LiDAR channels for application has not yet to be explored. While we designed the HSL to generate hyperspectral (continuous) data, we don’t know which channels are important. If we can select the optimal channels for different applications based on target spectral features, the HSL will be convenient for actual commercial applications.

Line 362: “channels typically”-> “channels that typically”

Reply: Fixed.

Line 369: “technique for hyperspectral LiDAR” so do you mean this can be as a postprocessing step? or on-board?

Reply: We are desire to develop channels select filter to select the useful channels for certain applications, and in this paper, we just make an software-based attempt to select optimal channels.

Line 371: “LiDAR can be simplified completely determined based” What do you mean?

Reply: Our mean is “Our proposed algorithm clearly demonstrates that few optimal hyperspectral LiDAR channels can obtain the desired results for targets classification application”. We combined this sentence with the previous sentence.

Line 372: “version AOTF-HSL for channels select filter”-> “version of AOTF-HSL for channels select filter” and what do you by the second part (channels select filter)

Reply: We rewrote the sentence. The filter in the second part is the acousto-optic tunable filter (AOTF), and AOTF is capable of filtering with a spectral band from 430 nm to 1450 nm at a high spectral resolution, therefore, AOTF offered a quicker tuning speed (microseconds) and broader wavelength ranges. We have designed an AOTF-HSL for experiments in a controlled laboratory environment, the filtered wavelength changes to generate consecutive and high-spectral-resolution laser pulses in time dimension, and the applicability of the system assessed proper range accuracy through the whole spectrum. While we designed the HSL, we didn’t know which channels are important, and we concern lower hyperspectral energy and miniaturized system is needed for on-site cases. So we are desired to select the optimal channels based on applications (such as target classification), and to design a miniaturized version AOTF-HSL with channels select filter based on application.

Line 375 and 376: change “form” to “from”

Reply: Fixed.

Reviewer 2 Report

The topic of the paper is very interesting and relates to current issues related to the construction of laser scanners operating in several or over a dozen channels. Multispectral and hyperspectral laser scanning is a hot topic in recent years and many papers describe its applications.

While reading the article, I had some general comments:

The title of paper is “Channels Selection Optimization for Hyperspectral LiDAR”. I would argue whether this title is appropriate, because in my opinion the methodology of channel selection does not lead to the construction of a hyperspectral laser scanner... Based on the definition of hyperspectral imaging, it seems that this is a methodology for the selection of channels for multi- or, at most, superspectral laser scanner construction for specific applications. So maybe the title of the paper should be rethought / improved.The more so, as the results of the research showed that the optimal MNSC is about 4 channels. I believe that this is a multispectral laser scanner, not an HSL. The authors propose a method that optimizes the selection of channels for the construction of a laser scanner, but for specific applications. However, because they used a limited number of samples (in each of the datasets) for analysis, the question arises whether the presented study will allow the selection of appropriate channels for the classification of various materials with high accuracy? Even more so in real conditions, not in the laboratory. I think, excessive minimization of the number of channels may prove to be inappropriate in practice. Do the authors suggest that for various application or different climate zones a specially dedicated laser scanner should be constructed? E.g. for the classification/detection of plant species? The random forest algorithm is also often used in the processing of hyperspectral data. Therefore, it would be worthwhile to perform analyzes for this and possibly other classification algorithms.

Detailed comments:

Line 17: what does it mean “information classification” in this sentence? I suggest to use “information extraction”. Lines 35-37: "8.5 channels", "4.5 channels", etc. - the number of channels should be an integer, so I'd suggest writing: ca. "8-9 channels", ca. “4-5 channels” etc. Lines 99-102: reference needed Lines 156: Why was only the 650-1100 nm range selected for analysis? Line 196: " namely goldenrain wood( Figure 1 (a)),gingko leaf" - spacing errors Equation 2: What does it means “i”? Line 258: should be “were” not “Were”. Table 3: specifying the average number of spectral channels in floating point values ​​is somewhat strange. I suggest writing “3-4” instead of "3.75".

Author Response

general comments:The title of paper is “Channels Selection Optimization for Hyperspectral LiDAR”. I would argue whether this title is appropriate, because in my opinion the methodology of channel selection does not lead to the construction of a hyperspectral laser scanner... Based on the definition of hyperspectral imaging, it seems that this is a methodology for the selection of channels for multi- or, at most, superspectral laser scanner construction for specific applications. So maybe the title of the paper should be rethought / improved.The more so, as the results of the research showed that the optimal MNSC is about 4 channels. I believe that this is a multispectral laser scanner, not an HSL. The authors propose a method that optimizes the selection of channels for the construction of a laser scanner, but for specific applications. However, because they used a limited number of samples (in each of the datasets) for analysis, the question arises whether the presented study will allow the selection of appropriate channels for the classification of various materials with high accuracy? Even more so in real conditions, not in the laboratory. I think, excessive minimization of the number of channels may prove to be inappropriate in practice. Do the authors suggest that for various application or different climate zones a specially dedicated laser scanner should be constructed? E.g. for the classification/detection of plant species? The random forest algorithm is also often used in the processing of hyperspectral data. Therefore, it would be worthwhile to perform analyzes for this and possibly other classification algorithms.

Reply: Compared with the hyperspectral imager, these HSLs had restricted and discrete spectral bands and channels, while a passive hyperspectral imager normally had hundreds of spectral bands with high spectral resolution. For broadening the application of HSL, special attention should be paid to develop the HSL with the continuous spectral band configuration and higher spectral resolution. While we designed the HSL to generate hyperspectral (continuous) data, we don’t know which channels are important. If we can select the optimal channels for different applications based on target spectral features. And our future work is designing a miniaturized version AOTF-HSL with channels adaptive filter based on applications.

We test four categories of materials and 27 kinds samples, not 27 samples, Our description is not exact, such as the rock/coal samples, not 4 samples, but 4 kinds samples. And the number of samples is limited, in the future research, we will collect more samples.

j: coal, k: rock from roof layer, l: rock from floor layer, m: gangue-rock

We prefer to test the newly developed system into a real application, however, as the hardware presented in following figure presents. The mobility of system should be necessarily improved. Thus in this study, we focus on channel selection, we hope to design a miniaturized version AOTF-HSL with channels select filter based on applications, and we believe that it is feasible.

We also explore the detection of plant species based on deep learning, this is the content of another paper we are researching. In this research, we test many classifiers, and we select two classifiers have the typical results, thank you for advice.

Detailed comments:

Line 17: what does it mean “information classification” in this sentence? I suggest to use “information extraction”.

Reply: Fixed. Thank you.

Lines 35-37: "8.5 channels", "4.5 channels", etc. - the number of channels should be an integer, so I'd suggest writing: ca. "8-9 channels", ca. “4-5 channels” etc.

Reply: Fixed. Thank you for suggestion.

Lines 99-102: reference needed.

Reply: Fixed.

Lines 156: Why was only the 650-1100 nm range selected for analysis?

Reply: The designed AOTF-HSL operates on a spectrum ranging from 650 nm to 1100 nm with 5nm average spectral resolution [details in Reference 25]. The laser emission unit consists of an SCL and an AOTF device, ensuring that different wavelengths of the laser beam can be emitted at each time slot. This design allows continuous wavelength selection of the laser pulse in the time dimension.

Line 196: "namely goldenrain wood( Figure 1 (a)),gingko leaf" - spacing errors.

Reply: Fixed.

Equation 2: What does it means “i”?

Reply: Thank you for mentioning. We added ‘ i =[1, 2,..., L], L denotes the total number of classes’ in this manuscript, and we modified Equation 2.

Line 258: should be “were” not “Were”.

Reply: Thank you. Fixed.

Table 3: specifying the average number of spectral channels in floating point values is somewhat strange. I suggest writing “3-4” instead of "3.75".

Reply: Fixed. Thank you for suggestion.

Round 2

Reviewer 1 Report

I thank authors for improving the quality of the manuscript, yet still, there are many grammatical problems in the manuscript and I strongly recommend the authors to ask a native speaker to polish the paper before resubmitting it again. I have listed some of these errors below:

Line 19: “widespread spectral information” -> “widespread spectral”

Line 30: “plant leaf with 27 kinds samples totally” -> “plant leaf with 27 samples totally”

Line 35-36: “We find that there are only 8-9 channels on average reached 100% accuracy by echo maximum and only 4-5 channels on average reached 100% accuracy by reflectance with the NB classifier” -> “to reach 100% accuracy, it demands only 8-9 channels on average by echo   maximum, while requiring only 4-5 channels on average when using reflectance based on NB classifier.”

Line 36: “while the” ->  ”these”

Line 38:  “much faster than randomly selected method” -> “much faster than the random selection method”

Line 39: “confirm the exacted channels” ->  do you mean “extracted”

Line 54: “applications. And the” -> “applications. The”

Line 57-58: “from the target on the selected wavelengths and the ranging” -> “from the targets at the
selected wavelengths as well as the ranging”

Line 60: “There are three kinds LiDAR” -> “There are three types of LiDAR”

Line 67: “operating on different” -> “operating at different”

Line 68: “different wavelength”-> “different wavelengths”

Line 84: “overing spectrum range from” -> “overing spectrum ranging from”

Line 85: “channel number of HSL”-> “channel numbers of HSL”

Line 86:  “AOFT [21].” -> “AOTF?? [21].”

Line 86: “designed 32-channel HSL” -> “designed a 32-channel HSL”

Line 89: “the wavelength of” -> “the wavelengths of”

Line 92: “we designed”-> write the name of authors here!!

Line 97: “also bring with” -> “also brings with”

Line 100 : “expand its application” you are talking about a specific application at the beginning of  this sentence!

Line 105: “have exploited” -> “have been exploited”

Line 108: “reduce calculate time” -> “reduce calculation time”

Line 114: “record in future”-> “records in future”

Line 115: “we don’t know” Please use the formal/academic terms

Line 128: “features extract from” -> “features extracted from”

Line 161: “27 kinds samples” -> “27 samples”

Line 163 (Table 2) : “Kinds of samples” -> “number of samples

Line 168: “four kinds of coal” -> “four types of coal”

Line 175: “after suitable dehydration, we collect” -> “after dehydration, where we collected”

Line 180: “ten species plant” -> “ten species of plant”

Line 210: “While we select improved classification” -> “If we select robust? classification”

Line 211: “parameters, the fewer channels are needed”  -> “parameters, the fewer channels will be needed”

Line 214-215: “correlation, it is necessary to confirm redundancy of the hyperspectral measurements is exist” -> UNCLEAR

Line 222: “standard variance,”-> “standard deviation?”

Line 222-223: “k is the number difference between the channels of two samples,” UNCLEAR

Line 224 (Figure 2): why you have 2 Figures of the same name (Figure 2). These  simple  things should be checked before submission

Line 225: “is shown in Figure 2” -> “Figure 3?”

Line 241: “two multiple classifiers” -> “two classifiers”

Line 248: “where the value is alterable, such as echo” -> “where value refers to echo”

Line 254: “is used as initial”  -> “is used as an initial”

Line 256: “channels is the optimum selected” -> “channels are the optimum selected”

Line 257 : “optimum select channels procedure” -> “optimum channel selection procedure”

Line 260: “Code for”-> “of the proposed?”

Line 260 : “in the paper)” -> “in this study”

Line 260;  step4: “correspond to” -> “corresponding to”

Line 260; output: “correspond to the selected” -> “corresponding to the selected”

Line 265: “as a feature.” -> “as the feature.”

Line 267: “of spectral channels as classification results, we defined it in reference” UNCLEAR

Line 268-269: “the classification of data is sufficient.” UNCLEAR

Line 270: “reached” -> “reaches”

Line 272: “and the MNSC with” -> “and with”

Line 277: “that the results of the” -> “that the”

Line 278: “algorithm decrease 38, 27, 53, 20 than” -> “algorithm selects 38, 27, 53, 20 bands less than”

Line 280: “drop by 8” -> “drop to 8”

Line 291: “classifier decrease 24, 4, 22, 12 than Normal” -> “classifier selects 24, 4, 22, 12 bands  less than Normal”

Line 292: “drop by 4, 3,” -> “drop to 4, 3,”

Line 293-295: “respectively. On average, there are only 4-5 channels that reach 100% accuracy with NB and only 2-3 channels that reach 100% accuracy with SVM.” UNCLEAR

Line 295: “classification with the proposed” -> “classification using the proposed”

Line 303-305: “The accuracy of the standard multiple classifiers, such as NB and SVM, could reach 100% while 3 or 4 channels are selected, and the channels could be provided specifically by the proposed algorithm.” -> “The accuracy of the standard classifiers, such as NB and SVM, could reach 100% while 3 or 4 channels were selected using the proposed algorithm.”

Line 306: “numbers by the proposed” -> “numbers using the proposed”

Line 308: “can calculate by” -> “can be calculated as”

Line 312: “features obtain” -> “the feature lead to”

Line 318 (Figurre 4): are the channel numbers  in nm?

Line 322: “from 670 nm to 1020 nm”-> does not match the range in the figure!!

Line 325: “4, the result coin with the previous” -> ? coin?

Line 333: “is smallest, therefore, the calculated time is least.” -> “is the smallest, therefore, the calculated time is minimal.”

Line 337: “based on 5 channels is 0.6043 seconds” -> I cannot make difference between this sentence and the previous one. the numbers are different, but you have not separate them in terms of the methods or whatever!

Line 343: “the channels selected applications to” -> what??

Line 344: “channels selected algorithm” -> “channel selection algorithm”

Line 353: “classification application” -> “classification applications”

Line 354: “version AOTF-HSL with channels select filter” -> “version of AOTF-HSL with a channel selection filter”

Line 357: “different directions. We”

Line 358: “will improve” -> “will also improve”

In the cove letter; 65: you have pointed out about the calibration of intensity to reflectance. it is recommended to explain how you have calibrated intensity values to reflectance!

Author Response

I thank authors for improving the quality of the manuscript, yet still, there are many grammatical problems in the manuscript and I strongly recommend the authors to ask a native speaker to polish the paper before resubmitting it again. I have listed some of these errors below:

Answer: Thank you for kinder suggestion. My co-authors polished the manuscript.

Line 19: “widespread spectral information” -> “widespread spectral”

Answer: Thank you for kinder suggestion. Fixed.

Line 30: “plant leaf with 27 kinds samples totally” -> “plant leaf with 27 samples totally”

Answer: Fixed, thank you.

Line 35-36: “We find that there are only 8-9 channels on average reached 100% accuracy by echo maximum and only 4-5 channels on average reached 100% accuracy by reflectance with the NB classifier” -> “to reach 100% accuracy, it demands only 8-9 channels on average by echo   maximum, while requiring only 4-5 channels on average when using reflectance based on NB classifier.”

Answer: Fixed.

Line 36: “while the” -> ”these”

Answer: Fixed.

Line 38:  “much faster than randomly selected method” -> “much faster than the random selection method”

Answer: Fixed.

Line 39: “confirm the exacted channels” ->  do you mean “extracted”

Answer: Fixed.

Line 54: “applications. And the” -> “applications. The”

Answer: Fixed.

Line 57-58: “from the target on the selected wavelengths and the ranging” -> “from the targets at theselected wavelengths as well as the ranging”

Answer: Fixed.

Line 60: “There are three kinds LiDAR” -> “There are three types of LiDAR”

Answer: Fixed.

Line 67: “operating on different” -> “operating at different”

Answer: Fixed.

Line 68: “different wavelength”-> “different wavelengths”

Answer: Fixed.

Line 84: “overing spectrum range from” -> “overing spectrum ranging from”

Answer: Fixed.

Line 85: “channel number of HSL”-> “channel numbers of HSL”

Answer: Fixed.

Line 86: “AOFT [21].” -> “AOTF?? [21].”

Answer: Fixed.

Line 86: “designed 32-channel HSL” -> “designed a 32-channel HSL”

Answer: Fixed.

Line 89: “the wavelength of” -> “the wavelengths of”

Answer: Fixed.

Line 92: “we designed”-> write the name of authors here!!

Answer: Fixed.

Line 97: “also bring with” -> “also brings with”

Answer: Fixed.

Line 100 : “expand its application” you are talking about a specific application at the beginning of this sentence!

Answer: I agree with your kinder suggestion, we rewrote the sentence.

Line 105: “have exploited” -> “have been exploited”

Answer: Fixed.

Line 108: “reduce calculate time” -> “reduce calculation time”

Answer: Fixed.

Line 114: “record in future”-> “records in future”

Answer: Fixed.

Line 115: “we don’t know” Please use the formal/academic terms

Answer: I agree with your kinder suggestion. We polished the sentence.

Line 128: “features extract from” -> “features extracted from”

Answer: Fixed.

Line 161: “27 kinds samples” -> “27 samples”

Answer: Fixed.

Line 163 (Table 2) : “Kinds of samples” -> “number of samples

Answer: Fixed

Line 168: “four kinds of coal” -> “four types of coal”

Answer: Fixed.

Line 175: “after suitable dehydration, we collect” -> “after dehydration, where we collected”

Answer: Fixed.

Line 180: “ten species plant” -> “ten species of plant”

Answer: Fixed.

Line 210: “While we select improved classification” -> “If we select robust? classification”

Answer: I agree with your kinder suggestion. We rewrote the sentence.

Line 211: “parameters, the fewer channels are needed”  -> “parameters, the fewer channels will be needed”

Answer: Fixed.

Line 214-215: “correlation, it is necessary to confirm redundancy of the hyperspectral measurements is exist” -> UNCLEAR

Answer: Thank you. We polished the sentence.

Line 222: “standard variance,”-> “standard deviation?”

Answer: I agree with your kinder suggestion. We polished the sentence.

Line 222-223: “k is the number difference between the channels of two samples,” UNCLEAR

Answer: We rewrote the sentence.

Line 224 (Figure 2): why you have 2 Figures of the same name (Figure 2). These simple things should be checked before submission

Answer: Thank you for suggestion.

Line 225: “is shown in Figure 2” -> “Figure 3?”

Answer: Fixed.

Line 241: “two multiple classifiers” -> “two classifiers”

Answer: Fixed.

Line 248: “where the value is alterable, such as echo” -> “where value refers to echo”

Answer: Fixed.

Line 254: “is used as initial”  -> “is used as an initial”

Answer: Fixed.

Line 256: “channels is the optimum selected” -> “channels are the optimum selected”

Answer: Fixed.

Line 257 : “optimum select channels procedure” -> “optimum channel selection procedure”

Answer: Fixed.

Line 260: “Code for”-> “of the proposed?”

Answer: Fixed.

Line 260 : “in the paper)” -> “in this study”

Answer: Fixed.

Line 260; step4: “correspond to” -> “corresponding to”

Answer: Fixed.

Line 260; output: “correspond to the selected” -> “corresponding to the selected”

Answer: Fixed.

Line 265: “as a feature.” -> “as the feature.”

Answer: Fixed.

Line 267: “of spectral channels as classification results, we defined it in reference” UNCLEAR

Answer: Thank you. We polished the sentence.

Line 268-269: “the classification of data is sufficient.” UNCLEAR

Answer: Thank you for mention. We polished the sentence.

Line 270: “reached” -> “reaches”

Answer: Fixed.

Line 272: “and the MNSC with” -> “and with”

Answer: Fixed.

Line 277: “that the results of the” -> “that the”

Answer: Fixed.

Line 278: “algorithm decrease 38, 27, 53, 20 than” -> “algorithm selects 38, 27, 53, 20 bands less than”

Answer: Fixed.

Line 280: “drop by 8” -> “drop to 8”

Answer: Fixed.

Line 291: “classifier decrease 24, 4, 22, 12 than Normal” -> “classifier selects 24, 4, 22, 12 bands less than Normal”

Answer: Fixed.

Line 292: “drop by 4, 3,” -> “drop to 4, 3,”

Answer: Fixed.

Line 293-295: “respectively. On average, there are only 4-5 channels that reach 100% accuracy with NB and only 2-3 channels that reach 100% accuracy with SVM.” UNCLEAR

Answer: Thank you. We polished the sentence.

Line 295: “classification with the proposed” -> “classification using the proposed”

Answer: Fixed.

Line 303-305: “The accuracy of the standard multiple classifiers, such as NB and SVM, could reach 100% while 3 or 4 channels are selected, and the channels could be provided specifically by the proposed algorithm.” -> “The accuracy of the standard classifiers, such as NB and SVM, could reach 100% while 3 or 4 channels were selected using the proposed algorithm.”

Answer: Fixed.

Line 306: “numbers by the proposed” -> “numbers using the proposed”

Answer: Fixed.

Line 308: “can calculate by” -> “can be calculated as”

Answer: Fixed.

Line 312: “features obtain” -> “the feature lead to”

Answer: Fixed.

Line 318 (Figure 4): are the channel numbers  in nm?

Answer: Figure 4 show the selected channels, we calculate the channel length with 650+5(n-1) nm.

Line 322: “from 670 nm to 1020 nm”-> does not match the range in the figure!!

Answer: Thank you for kinder suggestion, we correct ed the sentence.

Line 325: “4, the result coin with the previous” -> ? coin?

Answer: Thank you. We fixed it “is consistent with”.

Line 333: “is smallest, therefore, the calculated time is least.” -> “is the smallest, therefore, the calculated time is minimal.”

Answer: Fixed.

Line 337: “based on 5 channels is 0.6043 seconds” -> I cannot make difference between this sentence and the previous one. the numbers are different, but you have not separate them in terms of the methods or whatever!

Answer: I agree with your kinder suggestion. We polished the part.

Line 343: “the channels selected applications to” -> what??

Answer: Thank you, we rewrote this sentence.

Line 344: “channels selected algorithm” -> “channel selection algorithm”

Answer: Fixed.

Line 353: “classification application” -> “classification applications”

Answer: Fixed.

Line 354: “version AOTF-HSL with channels select filter” -> “version of AOTF-HSL with a channel selection filter”

Answer: Fixed.

Line 357: “different directions. We”

Answer: Fixed.

Line 358: “will improve” -> “will also improve”

Answer: Fixed.

In the cove letter; 65: you have pointed out about the calibration of intensity to reflectance. it is recommended to explain how you have calibrated intensity values to reflectance!

Answer: I agree with your kinder suggestion. Thank you.

Reviewer 2 Report

The authors gave comprehensive explanations of my doubts and questions. All my comments have been taken into account.

The manuscript has been improved and in its current version can be published in the Electronic journal.

Author Response

Thank you very much.